# Electrostatic and bending energies predict staggering and splaying in nonmuscle myosin II minifilaments

Tom L. Kaufmann[ORCID], Ulrich S. Schwarz[ORCID]*

Institute for Theoretical Physics and BioQuant, Heidelberg University, Heidelberg, Germany

* schwarz@thphys.uni-heidelberg.de

## Abstract

Recent experiments with super-resolution live cell microscopy revealed that nonmuscle myosin II minifilaments are much more dynamic than formerly appreciated, often showing plastic processes such as splitting, concatenation and stacking. Here we combine sequence information, electrostatics and elasticity theory to demonstrate that the parallel staggers at 14.3, 43.2 and 72 nm have a strong tendency to splay their heads away from the minifilament, thus potentially initiating the diverse processes seen in live cells. In contrast, the straight antiparallel stagger with an overlap of 43 nm is very stable and likely initiates minifilament nucleation. Using stochastic dynamics in a newly defined energy landscape, we predict that the optimal parallel staggers between the myosin rods are obtained by a trial-and-error process in which two rods attach and re-attach at different staggers by rolling and zipping motion. The experimentally observed staggers emerge as the configurations with the largest contact times. We find that contact times increase from isoforms C to B to A, that A-B-heterodimers are surprisingly stable and that myosin 18A should incorporate into mixed filaments with a small stagger. Our findings suggest that nonmuscle myosin II minifilaments in the cell are first formed by isoform A and then convert to mixed A-B-filaments, as observed experimentally.

**Data Availability Statement:** Our code and the required input data is available at GitHub as https://github.com/usschwarz/minifilaments. The data sets plotted in all figures are available at Zenodo as https://doi.org/10.5281/zenodo.3890896.

## Author summary

Nonmuscle myosin II (NM2) is a non-processive molecular motor that assembles into minifilaments with a typical size of 300 nm to generate force and motion in the actin cytoskeleton. This process is essential for many cellular processes such as adhesion, migration, division and mechanosensing. Due to their small size below the resolution limit, minifilaments are a challenge for imaging with traditional light microscopy. With the advent of super-resolution microscopy, however, it has become apparent that the formation of NM2-minifilaments is much more dynamic than formerly appreciated. Modelling the electrostatic interaction between the rigid rods of the myosin monomers has confirmed the main staggers observed in experiments, but cannot explain these high dynamics. Here we complement electrostatics by elasticity theory and stochastic dynamics to show that the parallel staggers are likely to splay away from the main axis of the minifilament and

**Funding:** USS is supported by the Deutsche Forschungsgemeinschaft (DFG, German Research Foundation) under Germany's Excellence Strategy through EXC 2181/1 – 390900948 (the Heidelberg STRUCTURES Excellence Cluster) and EXC 2082/1-390761711 (the Heidelberg-Karlsruhe Excellence Cluster 3DMM2O). He is also a member of the Interdisciplinary Center for Scientific Computing (IWR) at Heidelberg. The funders had no role in study design, data collection and analysis, decision to publish, or preparation of the manuscript.

**Competing interests:** The authors have declared that no competing interests exist.

that monomers attach and detach with rolling and zipping motions. Based on the sequences of the different NM2-isoforms, we predict that isoform A forms the most stable homofilaments and that A-B-heterofilaments are also very stable.

## Introduction

Myosin II is the most prominent subclass of the large myosin family of molecular motors that generate force and motion in the actin cytoskeleton [1]. Traditionally considered to be the force generator in muscle cells, with the rise of mechanobiology it has become clear that myosin II is also a central player in non-muscle cells [2]. Here it is not only involved in essential processes such as adhesion, migration and division, but also in mechanosensing. Regulated in cells by e.g. the Rho-pathway, nonmuscle myosin II (NM2) is assembled into minifilaments with approximately 30 single myosins [3, 4], forming a supra-molecular complex with a size around 300 nm that traditionally is investigated with electron microscopy [5, 6]. Only recently has live cell super-resolution microscopy made it possible to image the dynamics of NM2-minifilaments in cells [7–11]. This revealed many unexpected processes, including splitting, expansion, concatenation, long-range attraction and stacking. Moreover it was observed that single myosin monomers have a relatively high exchange rate [10] and that the three human NM2-isoforms A, B and C dynamically mix [12]. In a polarized cell, typically NM2A-minifilaments are formed at the front, then become mixed A-B-filaments as they move towards the back, and are mainly B-filaments at the back. Together, these recent observations lead to the central question how the molecular architecture of NM2-minifilaments, which formerly was assumed to be rather regular, can allow for such rich dynamics.

To address this important question, one has to investigate the physical basis of NM2-assembly. The two-headed NM2-monomer is actually a hexamer, with two identical heavy chains forming its backbone. Each of the two heavy chains consists of a globular and force-producing head at the N-terminus and a $\sim 160$ nm long $\alpha$-helical region which terminates with a short non-helical tailpiece at the C-terminus. The $\alpha$-helices of the two myosin heavy chains wind around each other due to interactions between periodically placed hydrophobic residues to form a so-called *coiled-coil*, which is an extremely rigid and regular structure with a pitch of 3.5 residues per turn and an axial spacing of 0.1456 nm between neighboring residues [13, 14]. While the coiled-coil myosin rod is held together by hydrophobic interactions, the assembly of these rods into minifilaments occurs mainly through electrostatic interactions between charged areas distributed along the tail. The prevalence of electrostatic interactions over other effects is demonstrated by the salt-dependence of the assembly process [15, 16]. While most of the NM2-rod has a net negative charge, there exists a small, highly-conserved region near the non-helical tailpiece of net positive charge which has been proven to be essential for filament assembly and is thus termed the *assembly critical domain* (ACD) [15, 17, 18]. This positive charge interacts mainly with five regions of increased negative charge distributed along the myosin rod [19]. Therefore electrostatics has to be the prime consideration to understand the architecture of NM2-minifilaments.

Due to a 196-residue charge repeat, favorable staggers between two parallel rods should be shifted at odd multiples of 98 amino acids with respect to each other [20, 21]. Electron microscopy and scattering experiments of myosin II and myosin II rod-fragments revealed prominent axial staggers of 14.3 nm, 43.2 nm and (less frequently observed) 72 nm between adjacent parallel myosin rods [22–24]. Recalling the axial spacing of 0.1456 nm between neighboring residues for the coiled-coil rod, these staggers indeed correspond to the odd multiples of 98

amino acids (14.3 nm / 0.1456 nm = 98.2, 43.2 nm / 0.1456 nm = 296.7 $\approx$ 3 · 98 and 72 nm / 0.1456 nm = 494.5 $\approx$ 5 · 98) and thus to the interactions between the positive C-terminal tip and the first three regions of increased net negativity. For antiparallel pairs, overlaps of 43–45 nm have been reported, which correspond to interactions between the positive C-terminal tip and the first region of increased net negativity. In its cross-section, the bare zone has a hexagonal structure with three rods for each side, giving rise to crowns of three myosin monomers of equal height. With five possible staggers on each side, this structure gives rise to $5 \times 2 \times 3 = 30$ monomers per minifilament, as observed experimentally [6, 25].

Previous modelling efforts have calculated the electrostatic interactions between the two rods based on the amino acid sequence of NM2 and their structure [16, 19, 24, 26]. While these electrostatic models have been able to explain the experimentally observed staggers as local minima of the electrostatic energy, this approach does not explain the rich dynamics recently observed in super-resolution live cell microscopy. Moreover, while the more simple variants of this approach produced relatively noisy results [19, 24, 26], the more detailed variant using the three-dimensional molecular structure of the coiled-coil found that globally almost all parallel staggers are energetically unfavorable [16]. This suggests that an essential element might has been missed and that the theory should be extended also into the time domain.

Here we propose an assembly model for NM2-minifilaments that not only considers the charge distribution as given by the sequence, but also the elastic energy arising from possible bending of the myosin heavy chains away from the main axis of the minifilament. Moreover we consider the process of dimer formation as a dynamic process in which the two rods align and de-align under thermal motion which we treat as a first passage time problem in an energy landscape including both electrostatics and elasticity. This new approach identifies the experimentally observed staggers as long-lived intermediates and makes new predictions for mixed filaments. Our results suggest that the contact time is a much more accurate measure of the stability of different configurations and that the bending and dynamic attachment of the rods are inherent features of NM2 dimer formation and minifilament dynamics.

## Materials and methods

### Structural model

The amino acid sequences of the myosin heavy chains were obtained from the *NCBI protein database* [27] with the accession numbers: NM2A—*P35579*, NM2B—*P35580*, NM2C—*NP_079005*, M18A—*Q92614*. The coiled-coil was translated into a linear chain of point charges with axial spacing of 0.1456 nm between residues [14]. The charges were assigned according to the charges of the amino acids (+2*e* for arginine and lysine and −2*e* for aspartic acid and glutamic acid; factor 2 as the rod domain consists of two heavy chains). We utilized the software *paircoil* [28] to determine the position of the individual amino acids within the rod and subsequently canceled the charges of amino acids that are facing towards the inside of the rod (i.e. positions *a* and *d* within the heptad repeat [13]). The rotational invariance of the model was justified with the findings from Ricketson et al. [16] who reported that even in their highly complex model, rotation around the main axis was insignificant for the final outcome. One issue that requires extra care is the treatment of the skip residues, which are single isolated residues outside the usual heptad structure of the coiled-coil that are believed to increase rod flexibility by locally perturbing the coiled-coil structure [29]. While the three skip residues of NM2 have been left out in the molecular model by Ricketson et al. [16], they have been kept in the charge chain model by Straussman et al. [19]. Here we follow the second approach because otherwise we would obtained an unrealistically high level of symmetry along the rod.

## Calculation of electrostatic interactions

In line with previous efforts, for straight NM2-rods we restrict our model to the electrostatic interactions [15, 16]. In principle, hydrophobic interactions could result from the local perturbations caused by the skip residues [29], but in general, it is well known that minifilament assembly is mainly determined by electrostatic interactions, as it is very sensitive to salt concentration. Electrostatics in the presence of mobile ions is a very challenging subject and in the strong coupling limit can lead to counterion condensation [30]. Assuming an effective charge of 0.085 e per residue (as obtained from the amino acid sequence of NM2B), a distance of 0.1456 nm between residues and modelling the myosin rod as a cylinder of radius 1 nm, we estimate a surface charge density of 0.09 $e/nm^2$ and subsequently a Gouy-Chapman length of 11 nm. Because this is much larger than the Bjerrum length of 0.7 nm, we are in the weak coupling limit described by Poisson-Boltzmann theory, a mean-field theory which assumes local thermal equilibrium [31, 32] and that has been used before to numerically calculate the electrostatic interactions of myosin rods based on a molecular model [16]. Here, however, we aim at a computationally more efficient model. A simple estimate shows that we are also in the limit of small ionic strengths (i.e. $q\Phi/k_BT \ll 1$). In this limit, the Poisson-Boltzmann equation can be linearized to yield the Debye-Hückel equation

$$\Delta\Phi(\vec{r}) = \kappa^2\phi(\vec{r}). \qquad (1)$$

For monovalent ions of concentration $n_0$, the range of the electrostatic interaction is given by the Debye-Hückel screening length $l_{DH}$ defined by

$$l_{DH}^2 = \frac{1}{\kappa^2} = \frac{\epsilon\epsilon_0 k_B T}{2e^2 n_0}. \qquad (2)$$

Here $\epsilon = 80$ is the dielectric constant of water. For the physiological salt-concentration $\sim 100$ $mM$ NaCl, the screening length takes the value $l_{DH} \approx 1.3$ nm [33]. Eq 1 is solved for a point particle with charge $Q$ by

$$\Phi(r) = \frac{Q}{4\pi\epsilon\epsilon_0}\frac{\exp(-\kappa r)}{r}. \qquad (3)$$

Due to the linear nature of Eq 1, the total electrostatic energy between two NM2-rods can be obtained by summation

$$E_{\text{total}} = \sum_i^N \sum_j^M \frac{q_i q_j}{4\pi\epsilon\epsilon_0}\frac{\exp(-\kappa r_{ij})}{r_{ij}} \qquad (4)$$

where $N$ and $M$ represent the total number of amino acids of the two rods and $i$ and $j$ the index of the respective rods.

## Bending of the rods

We treat the NM2-rods using the worm-like chain (WLC) model which describes the behavior of semi-flexible polymers, i.e. polymers with locally straight conformation. We assumed a persistence length of $l_p = 130$ nm [34, 35]. Bending was only considered for one of the two rods and restricted to a circular arc with radius $R$ and arc length $L_a$. The bending energy of the WLC then amounts to [32, 36]

$$E_{\text{bend}} = \frac{l_p k_B T L_a}{2R^2}. \qquad (5)$$

For a given stagger $s$, we assume that the myosin rod is straight along an overlap length $L_0$, then bends with radius $R$ and arc length $L_a$ and finally is straight again. In order to decrease the degrees of freedom, a grid-based minimization technique was used to fix $L_a$ and $R$ for given $s$ and $L_o$. This was realized by testing a range of possible values for $R$ and $L_a$ and calculating the electrostatic energy according to Eq 4 and the bending energy according to Eq 5. The ranges of tested values amount to $R \in [75 \text{ nm}, 300 \text{ nm}]$ and $L_a \in [15 \text{ nm}, 40 \text{ nm}]$. The minimum in total energy is used to fix $R$ and $L_a$ for given $s$ and $L_o$.

## Calculation of the contact time

We consider dimer formation as a two-step process. In the first step, the positively charged N-terminal region of the first rod attaches to the rod of the second NM2 with axial stagger $s$. We consider an initial contact of 25 residues which corresponds to 3.6 nm. Variations of the size of this initial contact had a negligible impact on the distribution of contact times. In the second step, the approaching NM2 would then roll onto the other NM2 to partially align the two rods. For the duration of this alignment process the stagger was considered to be fixed and detachment was only possible by reversing the attachment process. We modelled this alignment process as the diffusion of a Brownian particle in an external one-dimensional potential using the Fokker-Planck equation [37–39]

$$\partial_t p_2(x, t|x', t') = [-\partial_x A(x, t) + \partial_x^2 D(x, t)] p_2(x, t|x', t') \ . \tag{6}$$

The drift $A$ represents the directed motion of the particle and the diffusion constant $D$ the undirected Brownian motion. As we assume the stagger to be fixed, the free variable corresponds to the length of the overlap $x = L_o$ and the external potential $V(x) = V(L_o)$ to the energy potential with respect to $L_o$ for fixed $s$. The diffusion should be seen as an effective diffusion which describes the random alignment and de-alignment of the two rods. One would expect this effective diffusion to be dependent on the other variables (mainly $L_o$), however, due to the complex relationship between $L_o$ and $D$, we used a constant $D$ as a first approximation. In the overdamped limit we neglect the inertia of the system and thus, the directed motion of the particle directly stems from the external potential as $A(x) = \partial_x V(x)/\xi$ with the friction coefficient $\xi$. Furthermore, we assume constant diffusion as $D(x) = D = k_B T/\xi$ with the same friction coefficient $\xi$.

We are interested in the so-called *mean first passage time* $T_1$, i.e. the average time it takes the Brownian particle to leave the system domain $[a, b]$ defined by a reflecting boundary at $x = a$ (here $L_o = L_{\max}$) and an absorbing boundary at $x = b$ (here $L_o = 0$ nm). In our context, the mean time from the first contact to the separation of rods is the mean contact time. From our previous assumptions, the mean first passage time can be calculated as [37, 39]

$$T_1(x) = \frac{1}{D} \int_x^b dz \exp\left(\frac{V(z)}{k_B T}\right) \left[\int_a^z dy \exp\left(-\frac{V(y)}{k_B T}\right)\right] \ . \tag{7}$$

The contact time between two rods with respect to the initial attachment site (i.e. the axial stagger $s$ between the rods) poses a non-trivial transformation from the 2D potential energy to a measure of how likely the different staggers are, i.e. the contact time. In the absence of a microscopic model for the diffusion constant $D$, the contact times are not considered as absolute values but only in relation to each other.

## Results

### Staggering of myosin rods

Starting from the regular coiled-coil structure, we transformed the amino acid sequence into a linear chain of point charges with axial distance of 0.1456 nm between neighboring residues, similar to previous efforts [19, 24, 26] and as described in *Materials and Methods* (Fig 1A). Summing up all charges with a 98 residue window, we obtained the charge distribution along the myosin rod (Fig 1B). It clearly shows the known pattern of the positive end charge and the five negative charges distributed along the rod. Interestingly, the difference between the three different NM2-isoforms is relatively small. The electrostatic potential around the rod can be calculated using the Debye-Hückel equation (S1 Fig).

Next the two rods were placed next to each other with a stagger *s* (Fig 1C). The lateral distance between the rods was assumed to be 2 nm which is in accordance with previous research and results from the rod diameter of 1 nm [5]. We then calculated the overall electrostatic interaction energy (Fig 1D). We note that these results are very similar to the ones of a more detailed study that used a three-dimensional molecular structure of the coiled-coil [16], suggesting that our coarse-grained model captures the essential details of the charge distribution. For the parallel rods, the experimentally observed staggers at 14.3 nm, 43.2 nm and 72.0 nm emerged as local minima in the overall electrostatic energy landscape. For the antiparallel rods, the experimentally observed stagger at 113–115 nm (corresponding to an overlap of 43–45 nm with an overall rod length of 158 nm) emerged as the global minimum. However, as the exact length of the non-helical tailpiece is not known, the conversion from overlaps to staggers might not be completely accurate. In all cases, the observed staggers correspond mainly to interactions between the positively charged C-terminal end and regions of increased net negativity along the rod, but in addition also profits from complementary charges at other positions along the rod (S2 Fig). The interactions between antiparallel rods for negative staggers are extremely unfavorable as the positively charged ACDs are not in contact and therefore they are not considered in the following. For the favorable staggers, we find that the difference between the three different isoforms is relatively small (S3 Fig).

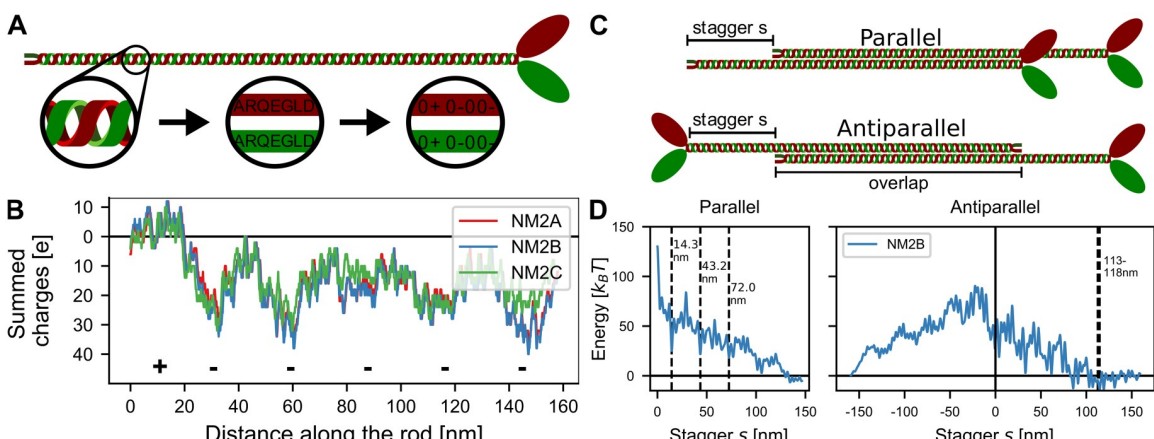

**Fig 1. Electrostatic interactions and staggering of myosin rods.** A: The myosin rod is treated as a linear chain of charges derived from the amino acid sequence of the respective isoform. B: Charge distribution along the rods. The charges were summed over a 98 residue window. The positive C-terminal tip as well as the five regions of increased net negativity are marked with + and -. All three isoforms are relatively similar. C: Schematic depiction of the parallel and antiparallel configurations with stagger *s*. D: Electrostatic energy of two straight NM2B rods as a function of stagger *s*. Dashed grey lines mark the experimentally observed staggers which agree with local energy minima (14.3 nm, 43.2 nm and 72.0 nm for parallel rods and 113–118 nm for antiparallel rods).

## Splaying of myosin rods

We next addressed the question which additional feature might be relevant to stabilize the metastable states identified by the electrostatic model. Since the myosin rods have a persistence length of $l_p \sim 130$ nm [34, 35], which is smaller than their length of 160 nm, appreciable bending is expected and indeed is commonly seen in electron microscopy images [6]. We therefore reasoned that bending might be an important aspect of minifilament stability. We considered the stagger $s$ and overlap $L_o$ to be free variables and to allow for variable bending after the overlap region (Fig 2A). For every set of $(s, L_o)$, we choose optimum values for the arc length $L_a$ and the radius $R$ to minimize the overall energy. The result of such a calculation is the energy as a function of both the stagger $s$ and the overlap $L_o$ (Fig 2B). The black diagonal line indicates the maximum possible overlap and the shaded area above this line represent inaccessible values for $L_o$. For parallel rods, it is clearly visible how the experimentally observed staggers at 14.3 nm and 43.2 nm can achieve larger overlaps $L_o$ (blue spikes in the heat plots), which indicates their high stability. The overall energies at these staggers are also more favorable (i.e. more attractive). For the antiparallel case, the experimentally observed staggers around $\sim 113$ nm are also visible, although several other favorable interactions are also conceivable. For small overlaps $L_o$, most of the staggers (in the parallel and antiparallel case) yield negative energies which reflects that the positive tip can attach at several negatively

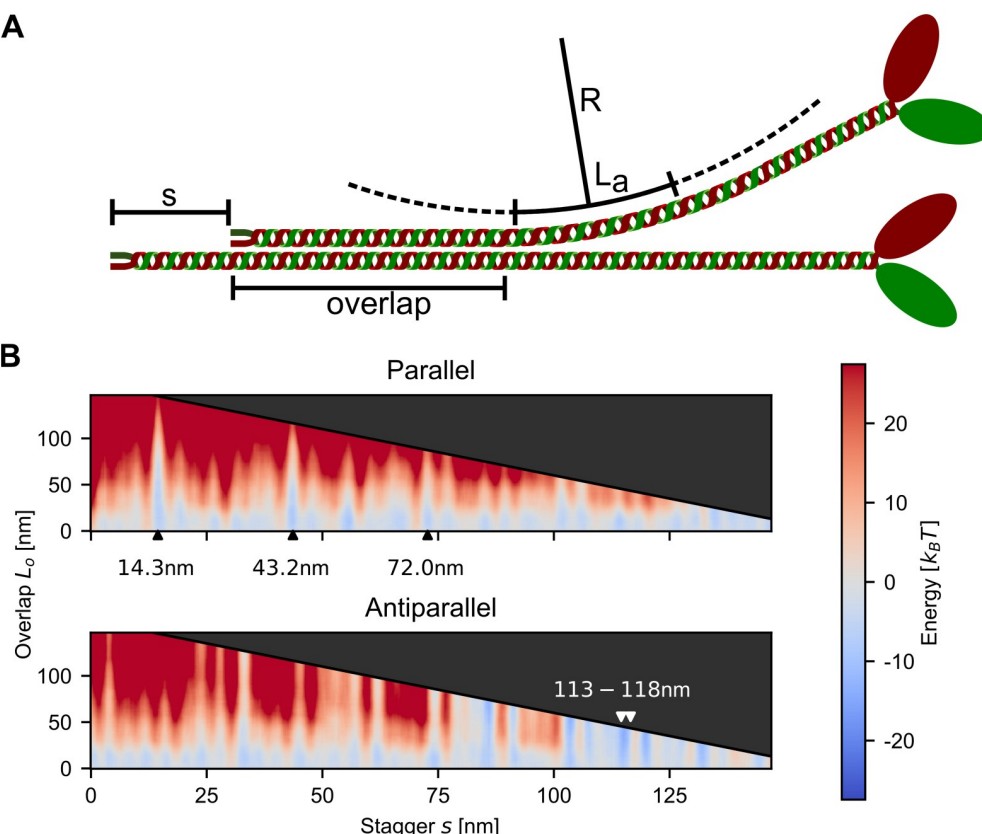

**Fig 2. Splaying of myosin rods.** A: Schematic depiction of the considered configurations. The two rods have axial stagger $s$ and overlap $L_o$. One of the two rods can bend along a circular arc with radius $R$ and arc length $L_a$. B: Total energy of two NM2B with respect to the stagger $s$ and the overlap $L_o$. The grey arrow heads indicate the experimentally observed staggers, which indeed correspond to favorable interaction energies (blue).

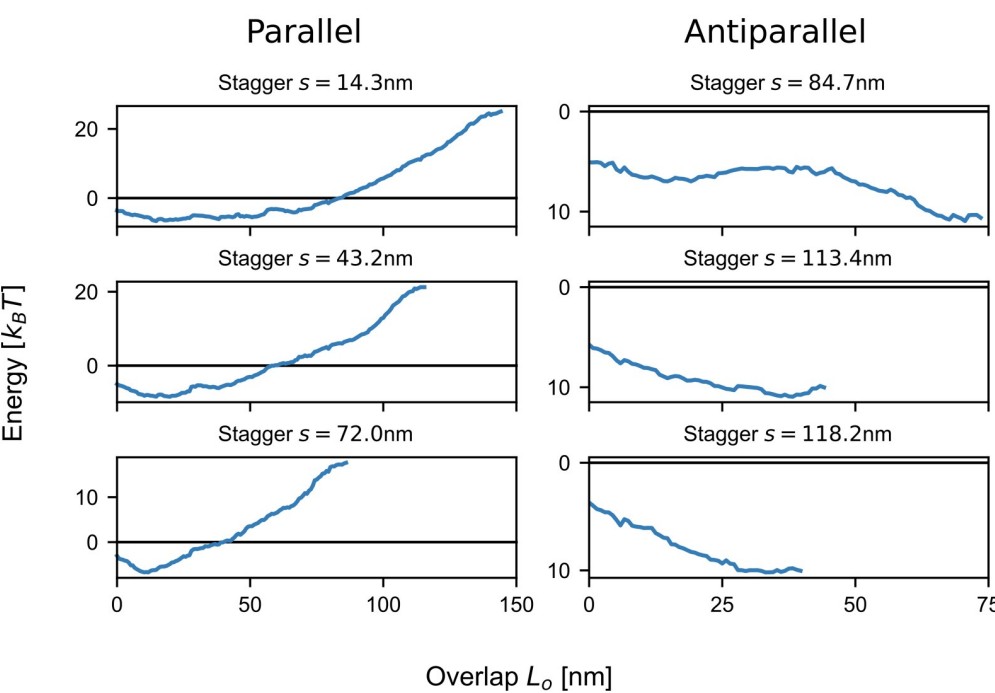

**Fig 3. Total energy of two NM2B as a function of overlap $L_o$ for fixed stagger $s$.** The parallel staggers have a critical overlap $L_o$ below which bending is favorable, while the antiparallel staggers exhibit decreasing overall energy for increasing overlap $L_o$. The minimum total energy for the parallel cases is between $\sim -6.5$ and $\sim -8k_BT$, while the antiparallel ones reach values of up to $\sim -11k_BT$.

charged sites along the rod, indicating a large diversity of possible configurations. As for the straight rods discussed above, the differences between the isoforms for the bent case are also relatively small (S4 Fig).

To better understand the relation between the total energy and the overlap $L_o$ for a given stagger $s$, we next plotted vertical cross-sections through the energy potential for NM2B (Fig 3). Similar results were obtained for the other two isoforms (S5 Fig). All parallel cases, which correspond to the experimentally observed staggers, are favorable for small overlaps and become unfavorable for larger overlaps. This strongly suggests that the formation of parallel dimers is always accompanied by the splaying of the heads away from the rods. By only partially aligning, the two rods prevent the negatively charged N-terminal end from interacting. For intermediate values of $L_o$, the potentials are rather flat with variations on the scale of $\leq k_BT$ for several tens of nm. This suggests that the actual value of $L_o$ fluctuates in the range of favorable values.

For the antiparallel case, the experimentally observed values at 113.4 nm and 118.2 nm as well as the stagger at 84.7 nm is shown. In contrast to the parallel case, the antiparallel staggers exhibit a steady decrease in overall energy with increasing overlap $L_o$, which is expected as both of the positively charged C-terminal regions can favorably interact on either side. This suggests that once attached at these staggers, the two antiparallel rods quickly align completely. The antiparallel staggers are also more tightly bound than the parallel ones. While the parallel cases achieve values between $\sim -6.5$ and $\sim -8\,k_BT$, the antiparallel ones have a maximum attraction of $\sim -11\,k_BT$. This suggests that the antiparallel configurations act as nucleation sites.

## Zipping and rolling of splayed rods

The splaying of myosin rods away from the minifilament main axis predicted above suggests strong interactions with the environment and high dynamics. In order to explore the potential dynamics of splayed configurations, we next treated the assembly process as a dynamical system subject to thermal motion. In detail, we imagined a two-step process of dimer-creation [26] (Fig 4A). In the first step, the positively charged tip of a NM2-rod is attracted to the large, negatively charged rod domain of another NM2-rod. After the tip attaches, the approaching NM2 would then roll onto the other NM2 to partially align the two rods. The length of the overlapping region between the two rods $L_o$ then stochastically fluctuates depending on the electrostatic potential. As binding energies between two rods reach several $k_B T$, the rods can only disassociate by unrolling, thus effectively reversing the attachment process. The stagger $s$ between the rods remains unchanged until the two rods disassociate.

In our dynamical framework, the quantity of main interest is the mean time for which two rods are engaged with each other, termed *mean contact time* $T_1$. Favorable staggers naturally result in higher mean contact times and therefore are more likely to be incorporated into minifilaments. We calculated $T_1$ by treating the alignment process as an effective Brownian motion in an external potential using the Fokker-Planck equation (Fig 4B). We note that the contact times scale inversely with the diffusion constant and are given in units of nm$^2$ because we abstain from parameterizing our simulations with a specific time scale as this would depend on context. We found that the contact time can reproduce the experimentally observed staggers for parallel rods (indicated by dashed lines in Fig 4B) much clearer than the total electrostatic energy shown in Fig 1D. These results suggest that the bending and dynamic attachment of the rods are inherent features of NM2 dimer formation and thus NM2-minifilament nucleation and growth.

## Mixed filaments

We next applied our contact time procedure to hetero-dimers, thus addressing the important question of mixed filament stability (Fig 5). The order of magnitude scaling factor on the top left of all diagrams reflects the strong variability in the contact times. The diagonal corresponds to the homo-dimers as discussed above. On the off-diagonals, the isoform that is listed first corresponds to the straight NM2-rod while the one listed second corresponds to the shifted and bent one. In the cell, the dimer formation between e.g. NM2A and NM2B is likely to be a mix between NM2A-NM2B and NM2B-NM2A as calculated here. For parallel interactions (Fig 5A), all pairings show strong peaks at the experimentally observed staggers of 14.3 nm, 43.2 nm and 72.0 nm. For combinations of NM2A and NM2B, the peak heights decrease, but the relative order stays the same. NM2C-containing dimers have their highest peak at $s = 43.2$ nm, as expected from the respective energy landscapes (S5 Fig). When comparing the overall contact times for parallel homo-dimers, NM2A has the highest contact time followed by NM2B and then NM2C. We furthermore find that NM2B-NM2A dimers have the highest peak of all pairings suggesting that mixed NM2B-NM2A dimers are the most stable. The contact times between NM2C and the other two isoforms are the lowest ones, which can be explained by the smaller net positivity of the C-terminal tip of the NM2C-rod.

The contact times for antiparallel interactions (Fig 5B) are about two orders of magnitude larger than the ones for parallel interactions, again strongly suggesting that antiparallel staggers act as nucleation centers for minifilament growth. Most pairings have their highest peaks at $s = 118$ nm, the same value as reported by Ricketson et al. [16] and close to the experimentally observed range (within the uncertainty of the non-helical tailpiece). As mentioned above, this stagger can be considered to be the interaction between the positive tip and the first region of

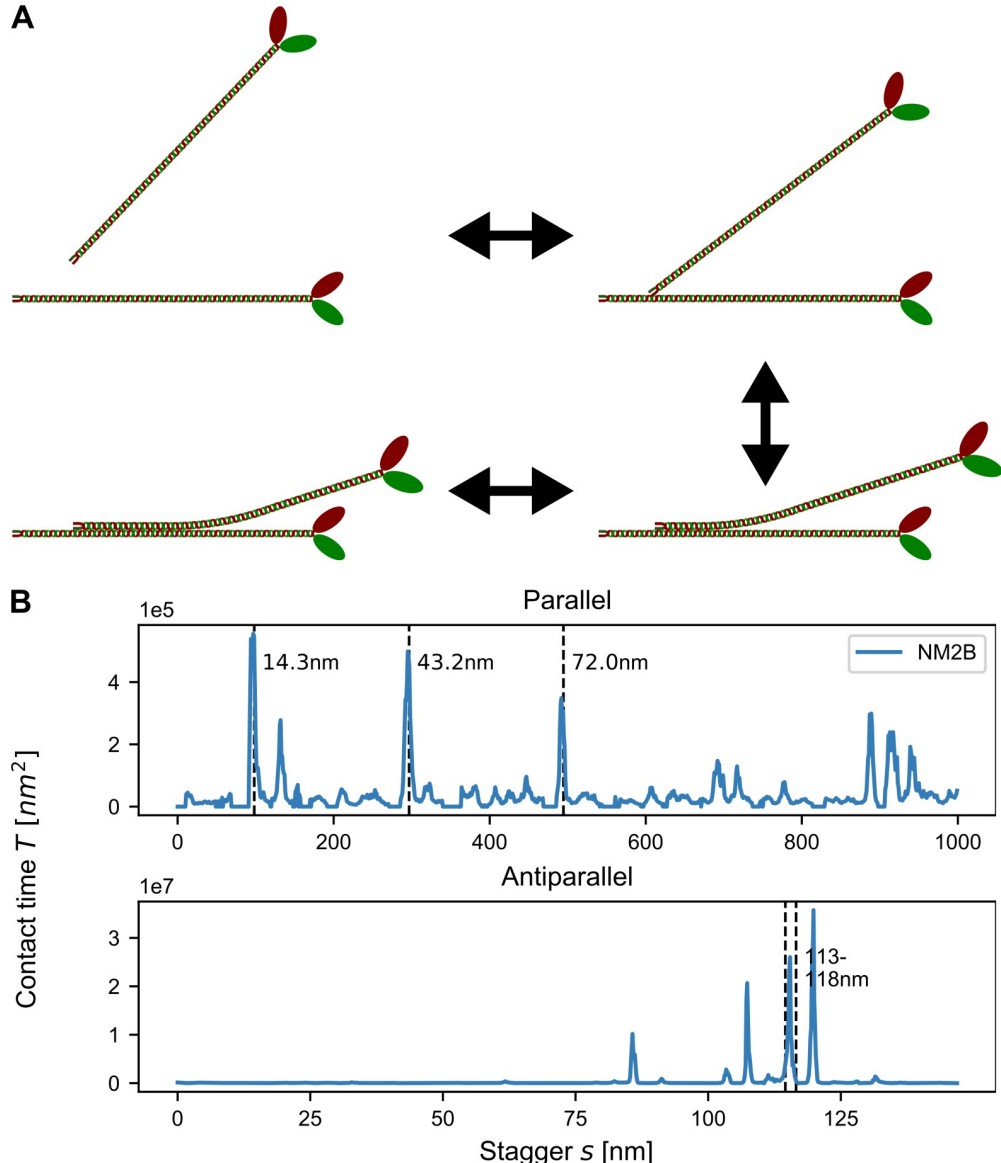

**Fig 4. Zipping and rolling of splayed rods.** A: Schematic depiction of the alignment process. B: Contact times calculated from the Fokker-Planck equation. Experimentally observed staggers (marked by the dashed grey lines) are present as clear peaks.

net negativity (S2 Fig) and corresponds to the maximum possible extension of two antiparallel rods. We note however that dimers containing NM2A have their highest peaks at $s = 85$ nm, which corresponds to the interaction between the positive C-terminal end and the second region of increased negativity (S2 Fig). The reason for this behavior might be that the first region of net negativity is weakest for NM2A. Similar to the parallel case, NM2A forms the longest lasting homo-dimers with contact times of roughly one order of magnitude larger than the NM2B and NM2C homo-dimers. NM2A-NM2B hetero-dimers are more stable than NM2B homo-dimers. In general, NM2A-NM2B have the highest number of pronounced peaks, while NM2C-containing hetero-dimers show the fewest ones. As the number of favorable configurations is likely linked to minifilament stability, these findings suggest that

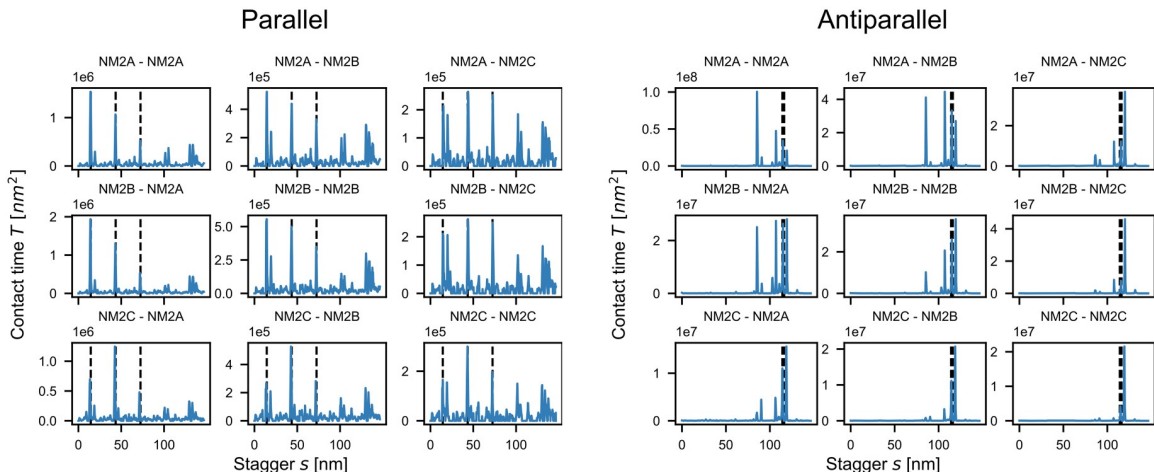

**Fig 5. Contact times for NM2 hetero-dimers.** The diagonal lines correspond to homo-dimers and the isoform that is mentioned first corresponds to the straight NM2-rod while the second one corresponds to the shifted and bent one. Experimentally observed parallel staggers (marked by the dashed grey lines) are recovered as stable configurations. Antiparallel configurations are more stable by two orders of magnitude. Surprisingly NM2A-NM2A homo-dimers have longer contact times than NM2B-NM2B homo-dimers.

NM2A-NM2B minifilaments are the most stable ones. Our results also agree with the observation that NM2C forms smaller minifilaments.

In order to extend the range of our analysis, we also tested the interactions between the NM2 isoforms and myosin 18A (M18A), a myosin that has been shown to be incorporated in small numbers into NM2-minifilaments [40]. For the case of parallel interactions, we found that there are fewer favorable staggers than between the isoforms with the longest-lasting parallel interaction close to the usual 14.3 nm stagger (S6 Fig). In the antiparallel case, the two major staggers around ∼85 nm and ∼114 nm are present. Our model thus shows why M18A can be incorporated into NM2-minifilaments. Furthermore, we predict the lack of favorable large parallel and small antiparallel staggers, which agrees with the observation that M18A assembles at the middle of the minifilaments [40].

## Discussion

Here we have presented a new model for the formation of NM2-minifilaments. Like most previous modelling efforts, we treated the NM2-rods as linear chains of charges interacting by electrostatics, but in addition we included the bending of the rods away from each other and treated the dimer formation as a dynamic process in which the two NM2-rods dynamically align and de-align. By employing the Fokker-Planck equation, we calculated the contact time for different staggers. Naturally, we would assume that configurations (i.e. staggers) with long contact times are more abundant in the cell and thus are more likely to be incorporated into minifilaments and to be experimentally observed. Indeed we found that the contact time between the rods presented a much more accurate measure for the quality of staggers than the electrostatic energy and identified the known staggers very well. More importantly, however, our results suggest that the bending and dynamic attachment of the rods are inherent features of NM2-dimer formation and thus NM2-minifilament nucleation and growth.

Following previous modelling efforts for the electrostatic interactions of the myosin rods, we have treated them as charged chains [19, 24, 26]. In contrast to these studies, however, we have not simply counted the number of opposing charges, but evaluated their interactions using Debye-Hückel potentials. Interestingly, we get very similar results for the shapes of the

electrostatic energy landscapes as the much more detailed model by Ricketson et al. using a three-dimensional molecular structure of the myosin rod [16] (compare Fig 1). We also obtain similar trends regarding the detailed interaction energies. Ricketson et al. have reported a maximal parallel repulsion of around 300 kcal/mole = 505.9 $k_BT$, a maximal antiparallel repulsion of around 250 kcal/mole = 422 $k_BT$ and a maximal antiparallel attraction of around 20 kcal/mole = 33.7 $k_BT$ (using $k_BT$ = 0.593 kcal/mole). We find considerably lower absolute values, but the same relative ordering, with maximal parallel repulsion at 120 $k_BT$, maximal antiparallel repulsion at 100 $k_BT$ and maximal antiparallel attraction of 10 $k_BT$ (see Fig 1D). We note that these lower values speak in favor of a more dynamic situation, including the fast exchange dynamics observed in experiments [10].

There are several possible explanations for the discrepancies in absolute values. Firstly Ricketson et al. might have used a different value for the ionic strength (not reported in their work). Secondly they deleted the skip residues from the sequence in order to be able to fit it onto a coiled-coil structure, thus increasing the regularity of the structure and the capacity for interaction. Because we are less interested in the absolute values of the electrostatic energies, but more in their relative values with respect to the staggers and how they translate into contact times, we conclude that the good agreement in terms of the relative binding energies strengthens our conclusions.

For the isoforms investigated here, we found that the favorable antiparallel configurations are more stable than the parallel ones by two orders of magnitude, which can be attributed both to the difference in total energy (antiparallel ones are $\sim 3$ $k_BT$ stronger, see Fig 3) as well as to the qualitative differences in their possible overlaps. For antiparallel interactions at staggers $\sim 118$ nm or 85 nm, the electrostatic energy between the rods is always favorable, even for the maximum possible overlap, and generally decreases for increasing $L_o$. In contrast, the parallel staggers at $s = 14.3$ nm or $s = 43.2$ nm become unfavorable for large overlaps $L_o$ and have an intermediate region with very flat energy potential which shows variations at the scale of $< k_BT$. From these potentials and the difference in contact times we expect that two antiparallel rods that get into contact at a stagger of $s = 118.2$ nm most likely completely align as the total energy becomes more favorable for larger overlaps. For two parallel rods that attach with the stagger $s = 14.3$ nm the overlap between the rods will likely fluctuate in the range $L_o \in$ [25nm, 75nm]. Together this means that while anti-parallel arrangements are expected to be straight, parallel ones are expected to be splayed away from the main axis.

Our results suggest that myosin II minifilaments use splaying and thermal fluctuations to spatially explore their environment for binding sites on actin filaments. In order to estimate the range of these explorations, we note again that the WLC-model leads to loss of correlation on the scale of the persistence length $l_p$ [32, 36]. Therefore, we assume that for a given stagger the splayed part of the rod is a circular arc of length $l_p$. For the first stagger at $s = 14.3$ nm with expected overlap $L_o \in$ [25nm, 75nm] (see Fig 3), the free part of the rod would be of length 85–135 nm (using a rod length of 160 nm). This leads to a maximal perpendicular distance of the heads to the backbone of 40–90 nm. As the expected values of $L_o$ are even smaller for the second and third staggers, even larger distances of the heads to the backbone are expected for these. Together we conclude that splaying should allow minifilaments to survey their environment for actin binding sites on a surprisingly large scale of several tens of nanometers. It is left to future work to explore this aspect further with a model of the actin environment of minifilaments.

Our procedures also allowed us to address the important question why the cell uses different NM2-isoforms and in particular mixed minifilaments. We found that for both parallel and antiparallel interactions, NM2A form the longest lasting homo-dimers, followed by NM2B and then NM2C. Moreover, we found that the NM2A-NM2B hetero-dimers are longer lasting

than the NM2B homo-dimers. These findings are surprising because NM2A is generally recognized to be the more dynamic of the two isoforms, while NM2B is regarded to be more stable. However, in the cell new minifilaments are mainly created at the leading edge by NM2A. These filaments then move backwards with the retrograde flow and NM2A is increasingly replaced by NM2B [2, 41, 42]. In this context, a higher contact time for NM2A-NM2A and NM2A-NM2B in fact seems very reasonable. The fact that NM2C has lower contact times as well as fewer favorable staggers than NM2A and NM2B agrees with the observation that NM2C forms smaller (and less frequent) minifilaments than its two counterparts.

For the parallel cases, our procedures nicely predict the experimentally observed staggers for all isoforms. For antiparallel rods, the results are more complicated. All isoforms show peaks around the experimentally observed overlaps of 43–45 nm (which corresponds to a stagger of 113–118 nm). As mentioned above, due to the unknown length of the non-helical tail piece, the experimentally observed overlaps and the staggers used here are not equivalent. Most antiparallel dimers also show a peak around $s \approx 84.7$ nm which corresponds to the interaction between the positively charged tip and the second region of increased net negativity (S2 Fig) and is therefore expected. However, it is not exactly clear why NM2A has its longest contact time at this stagger rather than at 113–118 nm. Dimers that contain NM2C generally show lower contact times for antiparallel staggers at $s \approx 84.7$ nm which would explain the larger bare-zone region and the smaller, overall size of NM2C minifilaments [6].

In the future, our approach can be also used to study the effect of mutations in the myosin rods. Although it cannot yet provide a full description of NM2-minifilament architecture, our results suggest the following global scheme (Fig 6). Due to the net negative charge distribution of the NM2-rod, an advancing NM2-rod will approach the other rod with its positively charged C-terminal tip first (Fig 6A). Once attached the two rods can align by rolling and zipping. However, the optimal stagger for the initial attachment is not well defined by the electrostatic energy alone as the initial attachment of the positively charged tip is favorable for almost all staggers. Therefore it seems likely that the two rods actually test different configurations by first attaching with the positively charged tip, stochastically aligning and de-aligning the two rods according to the energy potential and eventually detaching again. On average, the corresponding times should correspond to the contact times calculated above. Unfavorable staggers have low contact times, i.e. they separate quickly, while favorable staggers have long contact times. Obtaining the optimal stagger is therefore considered a trial-and-error process. The values of the contact times obtained here suggest that dimer formation in the cell is dominated by antiparallel dimers which build the starting point for minifilament nucleation (Fig 6B). The predominant mode of antiparallel interactions at $s \approx 114–118$ nm observed in this study corresponds to the maximum possible extension of a NM2-dimer and thus represents an ideal minifilament nucleation site as new rods can subsequently bind along the unobstructed regions of net negative charge. We note that it is this feature of minifilament nucleation that makes it so difficult to discern its exact organization in conventional light microscopy: additional growth steps will only add thickness, but not length to the complex.

Our model further suggests that rods which now attach in a parallel fashion only do so partially and bend away from the filament backbone (Fig 6C). We believe that splaying is an integral part of the formation of large-scale actomyosin structures as it enables the minifilament to search its vicinity for actin filaments and to bind several actin filaments at once. This ability to interact with a large number of surrounding actin filaments is required in particular for recently discovered processes such as filament splitting and templated nucleation [43] or the long-range attractions involved in stack formation [44]. Future advances in electron and super-resolution microscopy might make it possible to gather direct evidence during assembly for the splayed configurations as predicted here. It seems likely that the attachment of

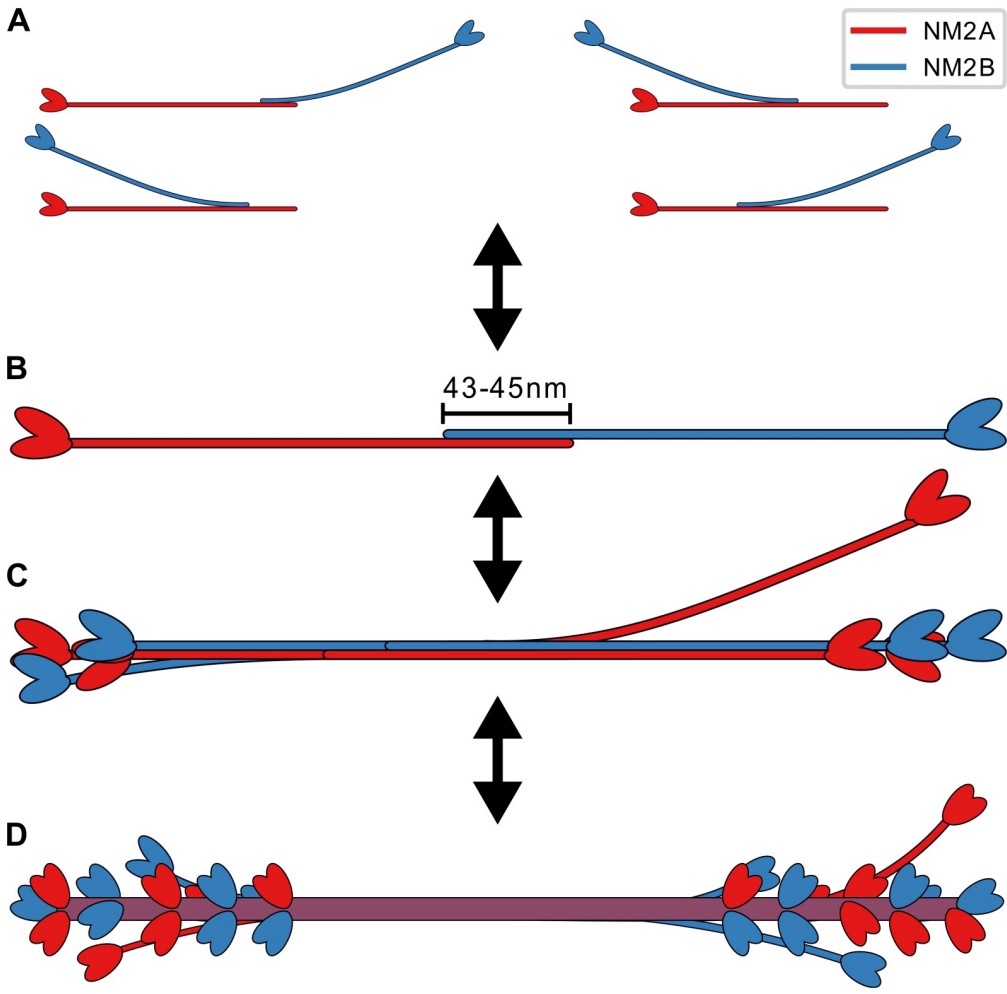

**Fig 6. Proposed scheme for mixed NM2-minifilament assembly.** A: Different homo- and hetero-staggers are tested by trial-and-error and with rolling motion. B: The stable antiparallel dimer with overlap of 43–45 nm serves as main scaffold for minifilament nucleation. C: Newly added parallel NM2 are splayed away from the backbone. D: The mature minifilament has five crowns of three monomers each on both sides totalling 30 monomers.

subsequent NM2-rods becomes more stable as the minifilament grows to reach its final size of roughly 30 monomers. This results from a mix of factors, including the stabilization of newly added NM2-rods by the interactions with the multiple rods already in the filament (Fig 6D). Yet even the fully assembled minifilament should be characterized by splayed myosin rods and a large degree of staggering disorder, which easily can interact with the environment and lead to the diverse processes recently observed in live cell microscopy. In the future, computer aided models should explore the interactions of several NM2 monomers to form larger oligomers. As the three-dimensional organization of single monomers in a growing minifilament is not yet fully understood, analytical models like the one presented here will not be sufficient and Brownian Dynamics or Molecular Dynamics simulations are required.

We finally note that myosin II minifilament assembly might also be shaped by the structural features related to the skip residues [19]. Despite being only three isolated residues inserted into the heptad repeat, skip residues are far more than punctual anomalies but rather extended regions of structural alterations, as shown recently for cardiac myosin II, which has four skip residues [29]. Skip residues lead to over- and underwinding of the coiled-coil and the first skip

residue has been shown to agree with the first of the two bends involved in the inactive and energy-saving form of myosin II [45]. Our approach of a regular linear chain of charges neglects potential changes in helical pitch and residue positioning due to the skip residues. In the future, molecular dynamics combined with appropriate coarse-graining strategies might be used to also address this important aspect of myosin II minifilament assembly. We speculate that the main effect will be increased flexibility of the myosin rods, thus further increasing the dynamic splaying effect that was modeled here with continuum theory.

## Supporting information

**S1 Fig. Electrostatic potential around the NMII-rod.** The NMII-rod was treated as a linear chain of charges and the electrostatic potential calculated using the Debye-Hückel theory (see Materials and methods). The subfigures show the electrostatic potential along the axis for variable distance radial distance $r$ = (top), $r$ = 2 nm (middle) and $r$ = 4 nm (bottom). The positively charged *ACD* as well as the five regions of increased net negativity are clearly visible for all three isoforms.
(TIF)

**S2 Fig. Experimentally observed staggers in relation to the charge distribution along the rod.** The charge distribution was calculated using a sliding window technique; the rod was treated as a linear chain of charges and the charges summed over a window of 98 charges. It is clearly visible how the experimentally observes staggers correspond to interactions between the positively charged ACD and the regions of increased net negativity.
(TIF)

**S3 Fig. Electrostatic interactions between straight rods.** While Fig 1D shows the results for NM2B, here we show them for all three isoforms. The three prominent parallel staggers and the one prominent antiparallel stagger are the same for all three isoforms.
(TIF)

**S4 Fig. Electrostatic interactions between bent rods.** While Fig 2B shows the results for NM2B, here we show them for all three isoforms. Again the same known staggers emerge for all three isoforms.
(TIF)

**S5 Fig. Total energy of two myosin rods as a function of overlap $L_o$ for fixed stagger $s$.** While Fig 3 shows the results for NM2B, here we show them for all three isoforms. These plots correspond to vertical cross-sections of S4 Fig. Again all isoforms show similar behavior.
(TIF)

**S6 Fig. Interactions between NMIIB and myosin 18A (M18A).** At the top we show a comparison of the charges along the respective rods using the sliding window technique (compare S1 Fig. The middle shows the total energy for parallel and antiparallel interactions. Here, the NMIIB remains straight while the M18A can bend away. The bottom plot shows the contact times between NMIIB and M18A with respect to the staggers. The most stable configurations are close to the experimentally observed values. There are no favorable large parallel or small antiparallel staggers, which is line with the experimental observation that M18A localizes at the middle of the minifilaments.
(TIF)

## Acknowledgments

We would like to thank Justin Grewe, Rasmus Schröder and Frauke Gräter for helpful discussions.

## Author Contributions

**Conceptualization:** Tom L. Kaufmann, Ulrich S. Schwarz.

**Investigation:** Tom L. Kaufmann.

**Methodology:** Tom L. Kaufmann, Ulrich S. Schwarz.

**Resources:** Ulrich S. Schwarz.

**Software:** Tom L. Kaufmann.

**Supervision:** Ulrich S. Schwarz.

**Visualization:** Tom L. Kaufmann.

**Writing – original draft:** Tom L. Kaufmann, Ulrich S. Schwarz.

**Writing – review & editing:** Tom L. Kaufmann, Ulrich S. Schwarz.

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
