## [Decision Letter · Decision Letter 0]

22 Apr 2020

Dear Prof. Dr. Schwarz,

Thank you very much for submitting your manuscript "Electrostatic and bending energies predict staggering and splaying in nonmuscle myosin II minifilaments" for consideration at PLOS Computational Biology. As with all papers reviewed by the journal, your manuscript was reviewed by members of the editorial board and by two independent reviewers. Based on the reviews and the guest editor's reading, we are likely to accept this manuscript for publication, providing that you modify the manuscript according to the review recommendations as well as to answer three questions by the guest editor listed below. 

Sincerely,

Dimitrios Vavylonis

Guest Editor

PLOS Computational Biology

Mark Alber

Deputy Editor

PLOS Computational Biology

[LINK]

The two reviewers of the manuscript as well as myself found your work important and novel. When you resubmit your manuscript, please consider their minor suggestions as well as the following questions I had:

1) Refs 16, 19, 24, 26 are mentioned as earlier studies where electrostatic energy calculations have been performed. Can you discuss any differences or similarities in terms of magnitude and overall trend (between prior and current energy calculations)? Would this comparison justify the level of coarseness used in this work?

2) In line 98 it is stated that “For small ionic strengths, the Poisson-Boltzmann equation can be linearized…” It’s not clear if this means that the linearization approximation is valid considering the stated charge density of the myosin tail. Can you comment on this approximation? Also, is the system far from regimes where counterions may cause attraction between rods of same charge?

3) Would accounting for the entropic fluctuations of the tail modify any of the results?

Reviewer's Responses to Questions

**Comments to the Authors:**

Reviewer #1: The authors described an insightful model that combines continuum electrostatics and elasticity to model the assembly of nonmuscle myosin II minifilament. By minimizing the effective energy of the system with different degrees of staggering and computing the relevant mean first passage time (referred to as "contact time"), the authors correctly predicted the favorable staggerings of parallel and anti-parallel dimers. Moreover, they demonstrated that the relatively simple model can also capture the differences between homodimers and heterodimers observed in recent experiments, supporting the general validity of the model.

Overall, the ms is clearly written with solid connection with experimental observations. Further extension of the model to multiple rods will be able to model more diverse behaviors for helping better interpret experimental observations. I support the publication of the work in PLoS Comp Biol.

I have only several minor questions/comments:

1. The model assumes a regular coiled-coil structure and therefore canonical charge distribution associated with such structure. In reality, however, there are skip residues that disrupt the regular coiled-coil pattern and modulate both the helical pitch and solvent-exposed residues (see, for example, recent work of I. Rayment, Cui and co-workers). The authors should discuss the potential impact of the skip residues.

2. Are there predictions that the authors can make for possible experimental tests?

Reviewer #2: This theoretical paper considers the configurations of parallel and antiparallel dimers of non-muscle myosin II molecules using the electrostatic and bending energies. Based on the distribution of charge along myosin rods the parallel dimers with staggers of 14.3, 43.2, and 72 nm present local energy minima. These staggers were previously predicted and observed experimentally. The novelty of this paper is that the authors consider that the length of contact between the two rods is not fully determined by the stagger, but could also change due to bending and splaying of the rods away from each other. Combined electrostatic and bending energy landscape identifies the same favorable staggers, but also suggests that parallel configurations have a strong tendency to splay myosin heads away from the common axis. In contrast, antiparallel configuration with an overlap of 43 nm is very stable and likely serves as nucleus for minifilament nucleation. Using Fokker-Planck equation, the authors compute relative contact times for different configurations and different myosin isoforms suggesting possible order of the assembly of myosin structures in the cell. This approach is novel and elegant, and has important implications for the understanding of the structure and dynamics of myosin II assemblies. I have only one minor comment: given the observations on rich myosin II structure and dynamics that are described in the Introduction, I would appreciate a little more discussion of the relevance of splaying of myosin heads to these variable configurations, e.g. parallel filament stacks.

**Have all data underlying the figures and results presented in the manuscript been provided?**

Reviewer #1: Yes

Reviewer #2: Yes

PLOS authors have the option to publish the peer review history of their article (what does this mean?). If published, this will include your full peer review and any attached files.

Reviewer #1: No

Reviewer #2: Yes: Alexander Verkhovsky
---

## [Decision Letter · Decision Letter 1]

28 May 2020

Dear Prof. Dr. Schwarz,

We are pleased to inform you that your manuscript 'Electrostatic and bending energies predict staggering and splaying in nonmuscle myosin II minifilaments' has been provisionally accepted for publication in PLOS Computational Biology.

Best regards,

Dimitrios Vavylonis

Guest Editor

PLOS Computational Biology

Mark Alber

Deputy Editor

PLOS Computational Biology

Reviewer's Responses to Questions

**Comments to the Authors:**

Guest Editor: Thanks for the detailed and clear explanations.

Reviewer #1: The authors have done a fine job responding to the questions raised in the last round of review.

**Have all data underlying the figures and results presented in the manuscript been provided?**

Reviewer #1: Yes

PLOS authors have the option to publish the peer review history of their article (what does this mean?). If published, this will include your full peer review and any attached files.

Reviewer #1: No

---

## [Editor Report · Acceptance letter]

26 Jun 2020

PCOMPBIOL-D-20-00418R1 

Electrostatic and bending energies predict staggering and splaying in nonmuscle myosin II minifilaments

Dear Dr Schwarz,

I am pleased to inform you that your manuscript has been formally accepted for publication in PLOS Computational Biology. Your manuscript is now with our production department and you will be notified of the publication date in due course.

With kind regards,

Laura Mallard
